# Sport Participation and Gender Differences in Dietary Preferences: A Cross-Sectional Study in Italian Adults

**DOI:** 10.3390/sports13080258

**Published:** 2025-08-06

**Authors:** Francesca Campoli, Elvira Padua, Michele Panzarino, Lucio Caprioli, Giuseppe Annino, Mauro Lombardo

**Affiliations:** 1Department of Human Sciences and Promotion of the Quality of Life, San Raffaele Open University, Via di Val Cannuta 247, 00166 Rome, Italy; francesca.campoli@uniroma5.it (F.C.); elvira.padua@uniroma5.it (E.P.); michele.panzarino@uniroma5.it (M.P.); 2Sports Engineering Laboratory, Department of Industrial Engineering, University of Rome Tor Vergata, 00133 Rome, Italy; lucio.caprioli@uniroma2.it; 3Accademia Nazionale di Cultura Sportiva, 00135 Rome, Italy; 4Human Performance Laboratory, Centre of Space Bio-Medicine, Department of Medicine Systems, University of Rome Tor Vergata, 00133 Rome, Italy; g_annino@hotmail.com

**Keywords:** sports participation, dietary preferences, gender differences, food choices, physical activity, cross-sectional study, Italian adults, nutritional habits, sport type and nutrition

## Abstract

Background: The relationship between sports participation and food preferences in adults, as well as the influence of gender, is still unclear. Objective: The objective of this study was to investigate the association between sports participation and individual food preferences and to explore potential gender differences among sports participants in a large group of Italian adults. Methods: This cross-sectional study involved 2665 adults (aged ≥ 18 years) who lead normal lives and underwent a routine lifestyle and dietary assessment at a clinical centre specialising in nutrition, metabolic health, and lifestyle counselling in Rome. Participants completed an online questionnaire on food preferences (19 foods) and sports practice. Multivariate logistic regression models, adjusted for age, sex, and smoking, were used to assess associations. Results: Sports participation was defined as engaging in structured physical activity at least once per week and was reported by 53.5% of subjects (men: 60.1%; women: 49.0%; *p* < 0.0001). After adjustment, active individuals were significantly more likely to prefer plant-based drinks, low-fat yoghurt, fish, cooked and raw vegetables, fruit, whole grains, tofu, and dark chocolate (all *p* < 0.05) and less likely to prefer cow’s milk (*p* = 0.018). Among sport participants, males were more likely to prefer meat (general, white, red, processed) and eggs, while females preferred plant-based drinks. No significant gender differences were observed for dairy products, legumes, or fish. Differences in food preferences were also observed according to the type of sport, with bodybuilders showing higher preference for tofu and dark chocolate. The strongest associations were found in the 25–44 age group. Conclusions: Sports participation is independently associated with specific food preferences, characterised by greater preference for plant-based and fibre-rich foods, and gender differences in food choices persist even among active adults. These findings highlight the need to consider both sports participation and gender when designing nutritional interventions.

## 1. Introduction

Physical activity and dietary habits are established determinants of health and play a key role in the prevention and management of chronic diseases, including obesity, type 2 diabetes, and cardiovascular disease [1]. Regular participation in sporting activities is associated with numerous health benefits, including improved metabolic profiles, reduced inflammation, and improved body composition [2,3]. However, the interaction between sports participation and dietary preferences remains poorly understood, especially in adult populations [4]. In addition, physically active individuals have specific nutritional needs, including increased energy and protein requirements to support muscle repair and recovery, increased carbohydrate intake for optimal performance, and adequate hydration and micronutrient intake to maintain health and prevent deficiencies [5]. Meeting these needs is essential not only for athletic performance but also for overall well-being and disease prevention.

Several studies suggest that individuals engaged in regular physical activity may adopt healthier dietary patterns, such as increased consumption of fruit, vegetables, and whole grains and reduced intake of processed foods and added sugars [6]. These behaviours may be influenced by both personal motivation and social factors, including gender roles and cultural expectations [7,8]. Food preferences, which reflect individual attitudes towards certain foods rather than actual consumption, are important determinants of eating behaviour and can influence both diet quality and athletic performance [9]. In addition, gender differences in both sports participation and food choices have been reported, with men and women showing distinct patterns in the types of sports played, frequency, and food choice [10,11].

Understanding the association between sport activity and food preferences, as well as potential gender differences, may provide useful insights into the development of tailored interventions to promote healthy lifestyles. However, data on the relationship between sport participation and food preferences in large, unselected adult populations remain scarce, and few studies have comprehensively examined the influence of gender on these associations [12,13].

This cross-sectional study aimed to examine the association between sports practice and individual food preferences in a large group of Italian adults. In particular, we sought to determine whether gender differences in food preferences persist among physically active individuals. Despite extensive evidence on the benefits of healthy eating and physical activity considered separately, little is known about how these behaviours interact, particularly with regard to the food preferences of active adults. To fill these gaps, the study sought to answer the following research questions: (1) Is sports practice associated with specific food preferences in adults? (2) Do physically active men and women differ in their food choices? We hypothesised that sports practice was linked to healthier food preferences and that gender differences persisted among sports practitioners.

## 2. Methods

### 2.1. Study Design and Participants

This cross-sectional study enrolled adults (aged 18 years or older) attending a medical centre specialising in nutrition and metabolic health in Rome, Italy, between May 2023 and June 2024 for routine evaluation. Inclusion criteria were as follows: age ≥ 18 years, residence in Italy, and completion of the entire questionnaire (including both sport participation and food preferences sections). Exclusion criteria were as follows: missing data on key demographic variables (age, gender), incomplete or inconsistent questionnaire responses, or self-reported diagnosis of chronic conditions that could significantly alter dietary habits (e.g., eating disorders, severe metabolic disease). All participants were community-dwelling adults without acute illness, hospitalisation, or diagnosed eating disorders at the time of participation. The sample size was not determined a priori using formal statistical software such as G*Power, therefore, no version number applies. Instead, all eligible adults attending the centre during the recruitment period were included. Of the 3302 initial participants recruited, a final sample of 2665 subjects was included, after excluding those who were underage, had not provided informed consent, had incomplete or inconsistent records, were affected by psychiatric disorders, pregnancy, or alcohol dependence, or lacked body composition data. The study was approved by the IRCCS San Raffaele Ethics Committee (protocol number RP 23/13) and registered on ClinicalTrials.gov (NCT06654674). Written informed consent was obtained from all participants.

### 2.2. Food Preference Data

Prior to the clinical evaluation, all participants completed an anonymous online questionnaire designed to collect comprehensive data on dietary habits and physical activity. The survey, accessible via any Internet-enabled device, was adapted from established food preference instruments for the Italian adult population, inspired by existing questionnaires used in Mediterranean dietary studies [14,15].

The food preference questionnaire included 19 foods relevant to the Mediterranean diet and commonly consumed by Italian adults: cow’s milk, plant-based drinks (e.g., soy milk), low-fat yoghurt, fresh cheese, meat in general, white meat, red meat, processed meat (e.g., ham), fish, eggs, legumes, cooked vegetables, raw vegetables, fruit, cereals (e.g., barley, spelt), whole grains, dried fruit, tofu, and dark chocolate (≥70%). For each food, participants answered “Yes”, “No”, “Reluctantly”, or “I don’t know” to the following question: “Do you like the following foods?”. Although this questionnaire has been used in previous studies by our group, it has not been formally validated; its content validity was assessed through a pilot phase with a subgroup of patients to ensure clarity and relevance. This limitation was taken into account in the interpretation of the results.

Both the dietary and physical activity data referred to current habits at the time of the first visit. The nutrition section included questions about food preferences, meal frequency, snacking habits, and context of meal consumption (e.g., alone or in company) and a brief 24 h dietary recall. However, for the present analysis, only data on food preferences were considered to maintain a clear focus on stable preference patterns.

No composite score or index was used; each food preference was analysed as a separate variable in the statistical models. “Meat in general” refers to the overall preference for meat, regardless of type. “White meat” included poultry (e.g., chicken, turkey); “red meat” included beef, lamb, and pork; and “processed meat” referred to cured products such as ham, salami, or sausages. The term “fresh cheeses” referred to low-fat products (e.g., ricotta, mozzarella, crescenza), while “natural yoghurt” referred to natural yoghurt with reduced sugar content. A further limitation is that preferences for beverages were not assessed, except for milk and plant-based drinks. We recognise that taste preferences may not directly correspond to habitual intake, which can be influenced by health, economic, or contextual factors. Data on 24 h dietary recall and other eating behaviours were collected but not included in the present analysis.

### 2.3. Tools for Assessing and Measuring Physical Activity

A dedicated section assessed physical activity. Participants indicated whether they regularly practised a sport or structured exercise (Yes/No), specifying the type of activity (e.g., bodybuilding, running, walking, swimming, total body, Pilates), frequency, and average hours per week. This approach allowed a detailed classification of sports participation and facilitated subgroup analyses by type and intensity of activity. The questionnaire also included items on meal frequency and eating and snacking habits and a brief 24 h diet recall. Weekly sport hours were assessed and categorised as <5, 5–10, or >10 h per week and were used both for descriptive analyses and as a covariate in regression models among sport participants.

### 2.4. Statistical Analysis

Descriptive statistics were calculated for all sociodemographic, lifestyle, and dietary variables, and the results were stratified by gender. Between-group comparisons for continuous variables were performed using independent t-tests (or one-way ANOVA for comparisons between multiple categories), while categorical variables were compared using the chi-square test. When assumptions of normality were not met, the Kruskal–Wallis test was used as an alternative. Bonferroni correction was applied for multiple comparisons where appropriate. Multivariable logistic regression analyses were conducted to explore the associations between each individual food preference (Yes/No) and sports participation (Yes/No), adjusting for age, gender, smoking status, and income. For each model, only subjects with complete data for the food in question, covariates, and outcome were included. The results are reported in the form of odds ratios (ORs) with 95% confidence intervals (CIs), β coefficients, standard errors (SEs), and *p*-values. Statistical significance was set at *p* < 0.05. All statistical analyses were performed with SPSS v. 28 (IBM Corp., Armonk, NY, USA). All figures were created using Python (v. 3.11) and its data visualisation libraries.

## 3. Results

Among the 2665 adults included in the analysis (1090 men and 1575 women), the mean age was 40.8 ± 13.3 years, with women significantly older than men (41.6 ± 13.4 vs. 39.5 ± 13.2 years, *p* < 0.001). No significant gender differences were found in the proportion of smokers (23.9% overall). The age distribution differed by gender, with more men in the 18–24 years group (13.5% vs. 9.8%, *p* = 0.004) and more women in the 45–64 years group (34.9% vs. 28.3%, *p* < 0.001). Annual household income was similar between sexes, except for the lowest income bracket, which was more frequent among women (< EUR 20,000: 17.7% vs. 13.7%, *p* = 0.006). Sports participation was reported by 53.5% of subjects, with a significantly higher proportion of men than women (60.1% vs. 49.0%, *p* < 0.001). Among sport participants, men were more likely to take part in bodybuilding (25.5% vs. 14.7%, *p* < 0.001) and running (9.9% vs. 5.0%, *p* = 0.001), while women were more likely to participate in walking (7.7% vs. 4.4%, *p* < 0.001) and total body exercise (4.1% vs. 2.5%, *p* = 0.001). No significant gender differences were observed for swimming or functional training. The weekly amount of sport activity was significantly higher in men, who more frequently reported 5–10 h or more than 10 h per week, whereas women more often reported less than 5 h per week (*p* < 0.001). The preferred time of day for sport was “Before dinner” (35.4%), with no significant differences between sexes in the distribution of preferred times of day. Full descriptive statistics are shown in Table 1.

As shown in Appendix A, significant differences were observed for several items, including cooked vegetables (92.3% in women vs. 84.5% in men, *p* < 0.0001), red meat (74.3% vs. 87.5%, *p* < 0.0001), and whole grains (81.3% vs. 73.1%, *p* < 0.0001), suggesting persistent gender-based food choices regardless of sport participation. Among the 19 foods assessed, individuals practising sport were significantly more likely to prefer vegetable-based drinks (*p* < 0.0001), low-fat yoghurt (*p* < 0.0001), whole grains (*p* < 0.0001), tofu (*p* < 0.0001), and dark chocolate (*p* = 0.0015). When stratified by sex within the group of sport practitioners, males reported significantly higher preferences for general meat (*p* = 0.0005), white meat (*p* = 0.0406), red meat (*p* = 0.0102), and legumes (*p* = 0.0017), whereas females more frequently preferred vegetable-based drinks (*p* = 0.0268), low-fat yoghurt (*p* = 0.0391), cooked vegetables (*p* = 0.0043), and nuts (*p* = 0.0200). No other comparisons of foods by gender were statistically significant (Figure 1).

Figure 2 illustrates the percentage of participants who answered “Yes” to specific foods, stratified by gender, age group (18–24, 25–44, 45–64, 65+), and sports participation. Only food × age × gender combinations are shown, with significant differences between practitioners and non-practitioners (*p* < 0.05). Among females, sports participants reported higher consumption of plant-based drinks, low-fat yoghurt, legumes, whole grains, tofu, and dark chocolate, especially in the 25–44 and 45–64 age groups. Among males, significant differences were observed in the same age groups for plant-based foods, legumes, and whole grains. The most consistent pattern between the two sexes was the greater preference for healthier foods among sports participants aged 25–44. Comparing genders within each age group, women generally showed higher percentages of “Yes” responses for vegetable-based drinks, legumes, cooked vegetables, and tofu than men, particularly among sports participants aged 25–44. Although the differences were not always statistically tested between genders in this figure, the visual trend suggests a more frequent adherence to health-conscious food choices among women.

Figure 3 compares the food preferences between the six most frequently reported types of sports: bodybuilding, running, walking, swimming, total body, and Pilates. The percentages represent the percentage of participants in each group who answered “Yes” to the consumption of each food. Statistically significant differences were observed for tofu (*p* = 0.0155), dark chocolate (*p* = 0.0356), and processed meat (*p* = 0.0378), with higher consumption of tofu and dark chocolate in the bodybuilding and total body exercise groups. No other foods showed significant differences between the various sport types (*p* > 0.05). The remaining comparisons reflect only descriptive trends.

Appendix A provides a comprehensive overview of food preferences by type of sport and gender. The heatmap shows the percentage of “Yes” answers for each food across the six most commonly practised sports. Although some visual differences can be seen between the various cells, the statistical analysis confirmed that significant gender differences emerged only among bodybuilding practitioners. In this group, females showed a higher intake of milk products, while males reported a higher consumption of red and processed meat. In all other types of sports, no statistically significant gender differences were observed.

Figure 4 presents a focused view of the statistically significant gender differences identified in the bodybuilding group. Females were more likely to consume low-fat yoghurt, cow’s milk, and white meat, whereas males showed a greater preference for red and processed meat. No significant gender differences were found in the other sports categories.

In the multivariate logistic regression analysis (Table 2), several individual dietary preferences were independently associated with sports participation after adjusting for age, gender, and smoking habits. Preferences for vegetable-based drinks, low-fat plain yoghurt, fish, cooked vegetables, raw vegetables, fruit, whole grains, tofu, and dark chocolate were all associated with a higher likelihood of participating in sports. Conversely, a preference for cow’s milk was associated with a lower likelihood of participating in sports. The strongest associations were observed for cooked vegetables (OR = 2.53; 95% CI: 1.40–4.57; *p* = 0.002), whole grains (OR = 1.96; 95% CI: 1.38–2.78; *p* < 0.001), and tofu (OR = 1.64; 95% CI: 1.33–2.03; *p* < 0.001). No significant associations were observed for other foods. These results confirm a trend towards healthier food preferences among physically active individuals in the sample.

Among individuals who reported regularly participating in sports, gender was a significant predictor of several specific food preferences (Table 3). Male participants were significantly more likely than women to prefer meat in general, white meat, red meat, processed meat, and eggs. In contrast, women reported a greater preference for vegetable-based beverages. The strongest associations were observed for red meat (OR = 3.17; 95% CI: 2.15–4.65; *p* < 0.001), meat in general (OR = 3.54; 95% CI: 2.00–6.28; *p* < 0.001), and processed meat (OR = 2.06; 95% CI: 1.49–2.86; *p* < 0.001). For most other foods, no significant differences were found between the sexes. These results indicate that significant gender differences in specific food preferences persist even among physically active adults. For the complete set of results, including all foods, see Appendix A.

A chi-square analysis comparing food preferences across four levels of weekly sport participation revealed significant differences for five food items: vegetable-based drinks, low-fat white yogurt, whole grains, raw vegetables, and tofu (all *p* < 0.001). As shown in Figure 5, the proportion of participants reporting a preference for these foods was generally higher among those engaging in sport, although not consistently increasing with the number of weekly hours.

## 4. Discussion

### 4.1. Association Between Sports Participation and Food Preferences

In this large cross-sectional study of Italian adults, we observed that specific food preferences are independently associated with participation in sporting activities. Active individuals were more likely to prefer vegetable-based beverages, low-fat yoghurt, fish, cooked and raw vegetables, fruit, whole grains, tofu, and dark chocolate, whereas cow’s milk was less frequently reported among those who played sports. These results provide new insights into the complex relationship between sports participation and dietary habits. This is in line with previous findings showing that physically active individuals tend to follow healthier dietary patterns, including a higher intake of plant-based foods and lower consumption of processed products [16,17]. Several cohort studies in Europe and North America have found that regular physical activity is associated with greater adherence to the Mediterranean diet or other health-oriented dietary patterns, even after adjustment for sociodemographic variables [18,19]. Our results confirm and extend these findings by showing a wider preference of sport participants for plant-based and fibre-rich foods. Recent evidence from Carey et al. [20] in a large cohort of Irish athletes and active individuals confirmed that both gender and sport type significantly influence food choices and that health awareness and performance-related factors are important determinants of diet in these populations. Similarly, Craddock et al. [21] highlighted a trend toward vegetarian-based dietary patterns in physically active individuals, often motivated by perceived health benefits and performance enhancement, with high overall diet quality but a potential risk of specific micronutrient deficiencies.

Interestingly, cow’s milk was found to be inversely associated with sports participation, contrary to international evidence from Anglo-Saxon countries, where milk is commonly consumed by athletes for muscle recovery [22,23]. This divergence could reflect cultural differences in Italy, where milk consumption by adults is less common and may be influenced by traditional Mediterranean patterns and a higher prevalence of lactose intolerance [24,25]. Such interpretations, however, deserve further investigation in longitudinal or qualitative studies. It is important to note that, in our sample, no significant gender differences were observed for cow’s milk preference.

### 4.2. Gender-Specific Dietary Patterns Among Sports Participants

Our data reveal marked gender differences in food preferences among athletes, with men showing a greater propensity to prefer meat (in general, white, red, and processed) and eggs, while women tend to prefer plant-based drinks. No significant gender differences were found for other foods, including fish, legumes, or cow’s milk. Although other studies have reported differences in dairy consumption between the sexes, we did not observe such differences in our sample. These results are consistent with previous research indicating that meat is more strongly associated with masculinity in Western cultures. For example, Rozin et al. [26] demonstrated, through several quantitative studies, the existence of a strong metaphorical link between mammalian muscle meat and masculinity, suggesting that social and cultural factors may underlie the persistent association between meat consumption and male identity.

A study by Rosenfeld and Tomiyama [27] further clarified that men not only consume more meat than women but are also less likely to adopt vegetarian diets. Their research shows that greater conformity to traditional gender roles among men predicts higher consumption of beef and chicken and less openness to vegetarianism, while among women, openness to vegetarianism is more often associated with health-related motivations. These findings are highly relevant in our context, where Italian male athletes reported a greater preference for meat (in general, white, red, and processed) and eggs, while female athletes showed a greater inclination towards plant-based drinks. Previous European studies have also shown that women are more likely to consider health aspects in their food choices and avoid certain foods for perceived health reasons, while men may be less restrictive and more likely to consume foods perceived as less healthy [28]. Furthermore, socioeconomic inequalities, cultural norms, and social distinction play a significant role in shaping food choices and physical activity patterns in both sexes [29]. Gender-based cultural norms can exert a strong and independent influence on dietary choices in the context of physical activity. This is consistent with our observation that the preference for meat among men was stable across all sports disciplines considered and supports the need for gender-sensitive approaches in dietary interventions targeting physically active populations.

### 4.3. Variation in Food Preferences by Type of Sport

Beyond sports participation in general, our results show that the type of sport was relevant to food preferences only for certain specific foods. Statistically significant differences between sports were found exclusively for tofu, dark chocolate, and processed meat, with higher consumption of tofu and dark chocolate among bodybuilding and total body exercise participants. No significant differences emerged for all other foods and sports.

This is in line with the current literature, according to which athletes who practise strength- or physique-focused sports often follow dietary strategies that favour high-protein foods or functional products aimed at supporting muscle growth, recovery, or changes in body composition [30,31]. Plant-based proteins, such as tofu, are becoming increasingly popular among male bodybuilders and athletes. As reported by van der Horst et al. [32], even in sports typically associated with traditional masculinity, plant-based diets are now more widely accepted and considered compatible with dedication to nutrition and training. This change reflects not only nutritional considerations but also the influence of professional nutritionists, online communities, and evolving social norms. Nutritional guidelines for physical athletes, as summarised by Roberts et al. [33], emphasise the importance of protein quantity and quality, carbohydrate management, and nutrient timing, with an increasing focus on the inclusion of plant-based sources in the diet.

Endurance athletes, on the other hand, tend to focus more on the overall quality of their diet and ensuring sufficient carbohydrate intake and are generally less likely to adopt highly restrictive or prescriptive regimes unless clearly necessary for performance or health reasons [34]. Social and psychological factors also play a role in determining dietary choices, especially among strength athletes, where social media, peer influence, and body image concerns are important factors [35,36]. The trend towards increased consumption of dark chocolate and tofu in bodybuilding may also reflect the popularity of “clean eating” approaches within the fitness community. Reviews in recent years have noted the growing role of plant-based proteins and meat substitutes as alternatives in sports nutrition, often linked to both health and ethical motivations [37]. Arora et al. [37] discuss the potential of meat substitutes in sport and the general population. When considering endurance and aerobic sports such as running and swimming, our data showed no substantial differences in dietary preferences compared to other active individuals. Similar results were reported by Huiberts et al. [38], who suggest that gender and training status should always be considered when evaluating the effects of different training modalities on diet and performance. Other factors, such as family environment and general food availability, also contribute to dietary choices, as discussed by leading national and international bodies [39]. Recent studies have shown that dietary habits and body satisfaction are closely related to physical activity and that gender differences are particularly evident in populations that engage in strength training [40]. Jimenez-Morcillo et al. [40] address this point, while West et al. [41] add important considerations for athletes following vegan diets.

Finally, the persistence of gender differences in specific types of sports, particularly bodybuilding, highlights the combined influence of sports culture and social norms. Research has shown that men who practise strength sports often prefer red meat and eggs for their perceived effects on muscle and performance, while women may prefer plant-based alternatives to support goals related to lean mass, aesthetics, or digestive well-being [35,36].

### 4.4. Age-Related Patterns in Dietary Preferences Among Sports Participants

The stratified analyses revealed that the association between sports participation and healthier food preferences was most pronounced among adults aged 25–44, while the differences were attenuated in the younger and older age groups. Similar results were reported by Wollmar et al. [42], who found that younger adults’ food choices are influenced by activity level, gender, and climate consciousness, with women more inclined toward plant-based and climate-conscious eating, while men and highly active individuals have greater energy needs and a higher carbon footprint. For the adult group, higher health awareness, responsibility for family dietary choices, and targeted nutrition education may contribute to the observed patterns [42,43]. Recent reviews further support that nutritional knowledge and lifestyle interventions in adulthood can shape long-term health behaviours [43,44]. In contrast, among younger adults, greater homogeneity in eating habits and limited autonomy in food choices may account for the absence of significant differences in food preferences between sport participants and non-participants [42]. For older adults, established dietary patterns, reduced variability in sport practice, or sample size limitations may explain the lack of clear associations [44]. Recent large-scale studies and reviews confirm that adherence to plant-based and health-oriented dietary patterns is linked to better cognitive, physical, and mental health in older age, supporting the importance of lifelong nutrition in healthy aging [45,46].

### 4.5. Strengths and Limitations

The main strengths of this study include the large sample size of adults attending a specialist medical centre, the detailed assessment of both physical activity and a wide range of dietary preferences, and the adjustment for major confounding factors. However, some limitations must be acknowledged. First, the dietary preferences questionnaire, although adapted from established tools used in Mediterranean diet studies, has not been formally validated. Its content validity was assessed only through a pilot trial with a subgroup of patients to ensure clarity and relevance. The absence of formal validation may have introduced measurement errors or biases, potentially limiting the accuracy and comparability of our results with those of studies using validated instruments. A further limitation is the exclusion of preferences for beverages, with the exception of milk and plant-based drinks, which were not systematically assessed in the questionnaire. This was a deliberate methodological choice to maintain the focus of the survey and reduce the burden on respondents, as well as based on evidence suggesting that solid foods contribute more substantially to dietary habits in this context. However, the omission of other beverages (such as sugary drinks, fruit juices, and alcohol) may have limited our ability to capture the full range of food preferences, and this should be taken into account when interpreting the results. The use of self-reported data for dietary preferences and physical activity may have introduced memory and social desirability biases, with participants possibly underestimating or overestimating certain foods perceived as more or less healthy. Although the questionnaire was administered anonymously and prior to clinical assessment to minimise bias, these limitations remain inherent in self-report methods. The cross-sectional design precludes any inference of causality between dietary preferences, physical activity, and associated outcomes. Unmeasured confounding factors, such as genetics, metabolic rate, or psychological factors (e.g., stress, body image concerns), may also have influenced the observed associations. Recruiting participants from a specialised medical centre in Rome may have resulted in a selection bias towards individuals with greater health motivation and may limit the generalisability of the results to other populations with different cultural or dietary backgrounds. Finally, although recruitment spanned nearly two years, no data were collected on seasonal variations in diet, which could influence the frequency of consumption of certain foods. Future research should consider longitudinal and multicentre designs, include validated tools for dietary and psychological assessment, and examine beverage preferences in more detail.

## 5. Conclusions

In conclusion, participation in sports activities is independently associated with specific food preferences, including a greater propensity to prefer plant-based drinks, low-fat plain yoghurt, cooked and raw vegetables, fruit, whole grains, tofu, dark chocolate, and fish and a lower preference for cow’s milk. Among sports practitioners, there are marked gender differences in food choices, with men more likely to prefer meat and eggs and women more likely to choose plant-based drinks. No significant gender differences were observed for dairy products, fish, or legumes. The type of sport influenced food preferences only for certain products, with higher preference for tofu and dark chocolate observed mainly among bodybuilders and total body trainers. The strongest associations between sports participation and healthy food preferences were observed in adults aged 25 to 44 years. These findings provide novel evidence on the interplay between sports engagement, gender, and dietary preferences in an Italian adult population.

## Figures and Tables

**Figure 1 sports-13-00258-f001:**
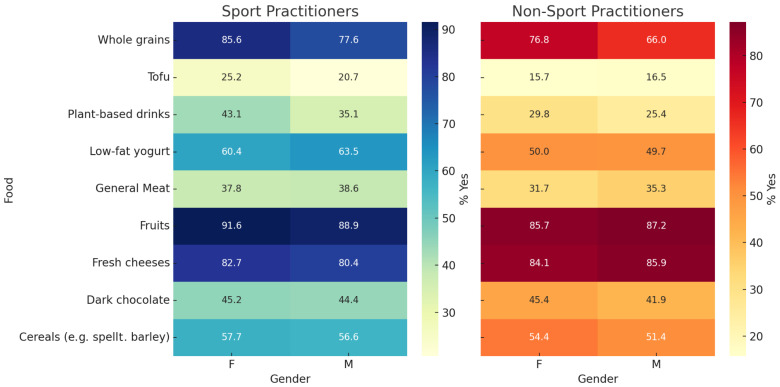
Gender differences in food preferences among sport and non-sport practitioners. This figure displays selected food categories from the questionnaire, stratified by gender and sport participation. Percentages represent the proportion of participants who responded “Yes” to each item. Two *p*-values are reported per food item: the first refers to the comparison between sport and non-sport practitioners and the second to the comparison between females and males among sport practitioners. The *p*-values are as follows: cow’s milk (*p* = 0.0932; *p* = 0.0891), plant-based drinks (*p* < 0.0001; *p* = 0.0268), low-fat yogurt (*p* < 0.0001; *p* = 0.0391), fresh cheeses (*p* = 0.0478; *p* = 0.6301), general meat (*p* = 0.0300; *p* = 0.0005), white meat (*p* = 0.3543; *p* = 0.0406), red meat (*p* = 0.0595; *p* = 0.0102), processed meat (*p* = 0.2953; *p* = 0.2157), fish (*p* = 0.3771; *p* = 0.0076), eggs (*p* = 0.2461; *p* = 0.3862), legumes (*p* = 0.0519; *p* = 0.0017), cooked vegetables (*p* = 0.1191; *p* = 0.0043), raw vegetables (*p* = 0.1609; *p* = 0.0745), fruits (*p* = 0.2695; *p* = 0.0683), cereals (*p* = 0.0010; *p* = 0.0970), whole grains (*p* < 0.0001; *p* = 0.1287), nuts (*p* = 0.1477; *p* = 0.0200), tofu (*p* < 0.0001; *p* = 0.1603), and dark chocolate (*p* = 0.0015; *p* = 0.4380). For the complete set of data, including non-significant comparisons, see Appendix A.

**Figure 2 sports-13-00258-f002:**
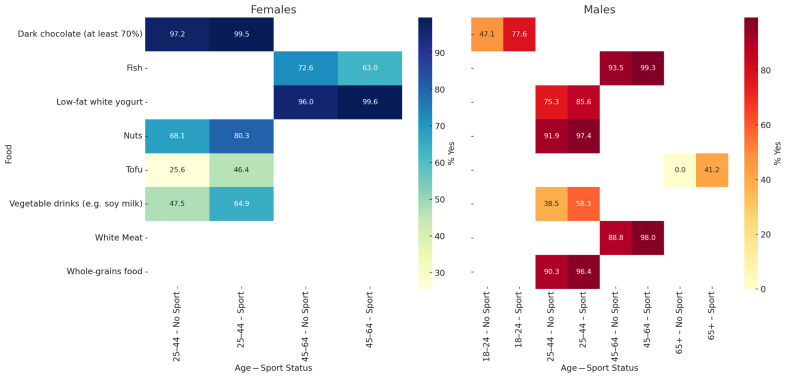
Significant differences in food preferences by gender, age group, and sport participation. These heatmaps show the percentage of participants who answered “Yes” for each food, stratified by gender (left: females; right: males), age group (18–24, 25–44, 45–64, 65+), and sports participation (Sport vs. No Sport). Only food × age × gender combinations showing statistically significant differences (*p* < 0.05) between those who participate in sport and those who do not were included. Female *p*-values (Sport vs. No Sport): cow’s milk (25–44: *p* = 0.0222), vegetable-based beverages (25–44: *p* = 0.0001; 45–64: *p* = 0.0012), low-fat yoghurt (25–44: *p* = 0.0029), general meat (45–64: *p* = 0.0046), red meat (45–64: *p* = 0.0077), pulses (25–44: *p* = 0.0077), legumes (25–44: *p* = 0.0016), cooked vegetables (25–44: *p* = 0.0023), cereals (45–64: *p* = 0.0036), whole grains (25–44: *p* < 0.0001), tofu (25–44: *p* = 0.0063), and dark chocolate (25–44: *p* = 0.0188). Male *p*-values (Sport vs. No Sport): cow’s milk (18–24: *p* = 0.0436), vegetable-based beverages (25–44: *p* = 0.0002), low-fat yoghurt (25–44: *p* < 0.0001), red meat (25–44: *p* = 0.0122), pulses (25–44: *p* = 0.0126), cooked vegetables (25–44: *p* = 0.0088), cooked vegetables (25–44: *p* = 0.0039), raw vegetables (25–44: *p* = 0.0137), whole grains (25–44: *p* < 0.0001), tofu (25–44: *p* = 0.0077), and dark chocolate (25–44: *p* = 0.0100).

**Figure 3 sports-13-00258-f003:**
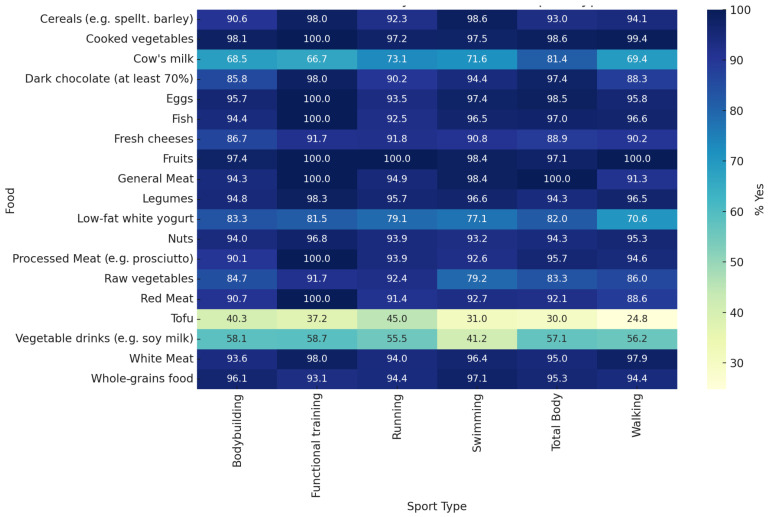
Dietary preferences across six most practiced sport types. This heatmap shows the percentage of participants who answered “Yes” to 19 food items, stratified by the six most commonly reported types of sports: bodybuilding, running, walking, swimming, total body, and Pilates. Food preferences were collected from the responses for the primary and secondary sport types. The following *p*-values reflect the comparisons between the six sport groups for each food: cow’s milk (*p* = 0.3825), vegetable beverages (*p* = 0.2890), low-fat yoghurt (*p* = 0.0601), fresh cheese (*p* = 0.1585), general meat (*p* = 0.5214), white meat (*p* = 0.6135), red meat (*p* = 0.5716), processed meat (*p* = 0.0378), fish (*p* = 0.2794), eggs (*p* = 0.0378), eggs (*p* = 0.4713), legumes (*p* = 0.1300), cooked vegetables (*p* = 0.1327), raw vegetables (*p* = 0.0754), fruits (*p* = 0.2767), cereals (*p* = 0.4766), whole grains (*p* = 0.4851), nuts (*p* = 0.6422), tofu (*p* = 0.0155), and dark chocolate (*p* = 0.0356). Items marked with an asterisk (*) indicate statistically significant differences in food preferences across sport types (*p* < 0.05).

**Figure 4 sports-13-00258-f004:**
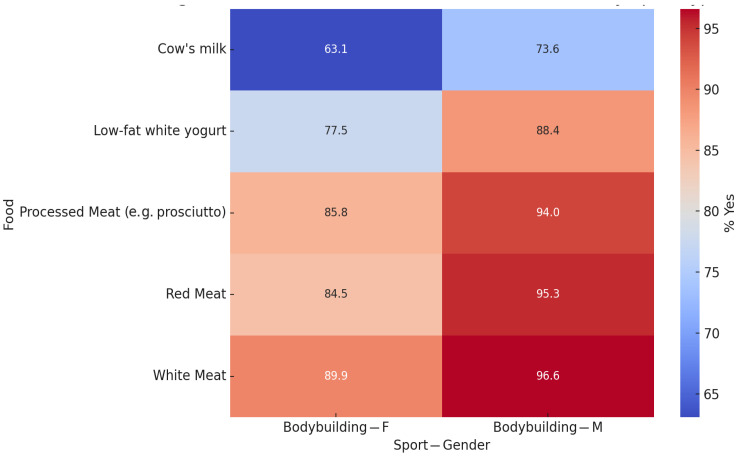
Significant gender differences in food preferences by type of sport. This heatmap shows the food preferences (% of “Yes” responses) of male and female participants within the six most commonly reported sport types: bodybuilding, running, walking, swimming, total body, and Pilates. Only food–sport combinations with statistically significant gender differences (*p* < 0.05) are shown. Significant differences were only observed in the bodybuilding group for processed meat (*p* = 0.0018), low-fat yoghurt (*p* = 0.0023), red meat (*p* = 0.0026), cow’s milk (*p* = 0.0150), and white meat (*p* = 0.0317). No other sports groups showed significant variations according to gender. Blank cells were excluded from the analysis due to missing data.

**Figure 5 sports-13-00258-f005:**
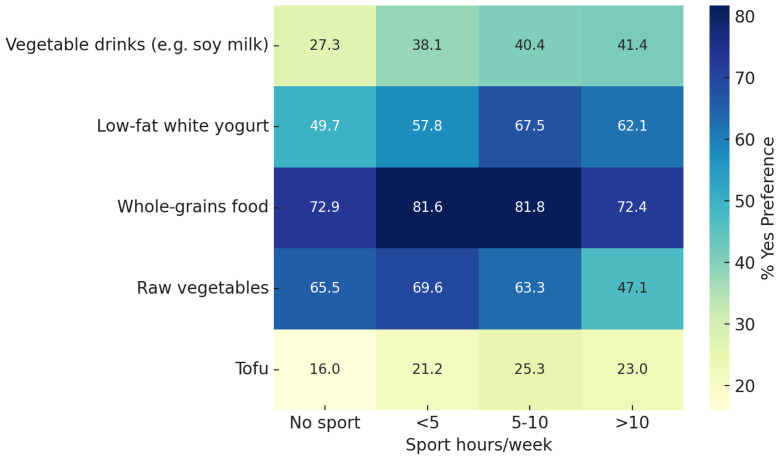
Significant gender differences in food preferences by type of sport. Heatmap showing the percentage of participants who reported a positive preference (“Yes”) for selected food items, stratified by weekly hours of sport practice. Significant differences across sport activity levels were observed for the following items (chi-square test): vegetable-based drinks (*p* = 4.7 × 10^−6^), low-fat white yogurt (*p* = 7.5 × 10^−7^), whole grains (*p* = 4.0 × 10^−5^), raw vegetables (*p* = 1.5 × 10^−5^), and tofu (*p* = 2.5 × 10^−6^).

**Table 1 sports-13-00258-t001:** Descriptive characteristics of the study population overall and stratified by gender. Table 1 shows the main demographic and behavioural characteristics of the study sample, stratified by gender (*n* = 2665). Values are expressed as means ± standard deviations for continuous variables and percentages for categorical variables. *p*-values are derived from chi-square tests (categorical) or *t*-tests (continuous). Statistically significant gender differences were observed for age, sports participation, specific types of sports (bodybuilding, running, walking, total body), and low income level (<EUR 20,000/year). For the subgroup of sport participants, the distribution of weekly physical activity hours and preferred time of day is also shown, with *p*-values for the gender comparison. All *p*-values are shown in full; for significant results with *p* < 0.0001, the value is indicated as such.

			Total (*n*)	M	F	*p*-Value
	Subjects	*n*	2665	1090	1575	–
	Smokers	yes	23.9%	23.3%	24.4%	0.5173
Age	Age	yrs	40.8 ± 13.3	39.5 ± 13.2	41.6 ± 13.4	<0.0001
	18–24	%	11.3%	13.5%	9.8%	0.0043
	25–44		51.3%	53.8%	49.5%	0.0347
	45–64		32.2%	28.3%	34.9%	0.0003
	65+		5.2%	4.5%	5.7%	0.1927
INCOME PER YEAR	<EUR 20,000	%	16.1%	13.7%	17.7%	0.0061
	EUR 20,000—EUR 40,000		67.8%	69.0%	67.0%	0.2947
	EUR 40,000–EUR 60,000		13.1%	14.2%	12.3%	0.1548
	>EUR 60,000		2.8%	2.9%	2.7%	0.8443
	No participation in sport	%	46.4%	39.9%	50.9%	<0.0001
	Participation in sport		53.5%	60.1%	49.0%	<0.0001
	Bodybuilding	%	19.1%	25.5%	14.7%	<0.0001
	Running		7.0%	9.9%	5.0%	0.0012
	Walking		6.3%	4.4%	7.7%	<0.0001
	Swimming		4.5%	5.9%	3.5%	0.1245
	Total body		3.5%	2.5%	4.1%	0.0010
	Functional training		2.7%	3.2%	2.3%	0.7269
	Variable	%	Total (*n* = 1415)	Males (*n* = 655)	Females (*n* = 760)	*p*-value
	<5 h/week sport	%	49.7	44.9	53.8	<0.0001
	5–10 h/week sport	%	38.4	42.2	35.2	
	>10 h/week sport	%	11.9	12.9	11.0	
	Most frequent time of day for sport		Total (*n* = 1355)	Males (*n* = 621)	Females (*n* = 734)	*p*-value
	Before Breakfast	%	6.7	7.3	6.2	0.4955
	After breakfast	%	9.3	8.2	10.2	
	During the morning	%	17.1	15.9	18.1	
	In the afternoon	%	25.1	26.4	24	
	Before dinner	%	35.4	35.7	35.1	
	After dinner	%	2.2	2.4	1.9	

**Table 2 sports-13-00258-t002:** Associations between individual food preferences and sports participation in Italian adults. Multivariable logistic regression models evaluating the association between sports participation (dependent variable, Yes/No) and each food preference (independent variable, Yes/No), adjusted for age, gender, and smoking status. For each food, the table reports the regression coefficient (β), standard error (SE), 95% confidence interval (CI) for β, odds ratio (OR), 95% CI for OR, and *p*-value. All 19 food preferences are included.

Food Item	β	SE	95% CI (β)	OR	95% CI (OR)	*p*-Value
Cow’s milk	−0.19	0.08	−0.35, −0.03	0.82	0.70, 0.97	0.018
Vegetable drinks (e.g., soy milk)	0.45	0.08	0.28, 0.61	1.56	1.33, 1.84	<0.001
Low-fat white yogurt	0.42	0.08	0.27, 0.58	1.53	1.31, 1.79	<0.001
Fresh cheeses	−0.06	0.11	−0.27, 0.15	0.94	0.76, 1.16	0.570
General meat	0.29	0.18	−0.05, 0.63	1.34	0.95, 1.89	0.097
White meat	0.16	0.15	−0.14, 0.45	1.17	0.87, 1.57	0.293
Red meat	−0.05	0.13	−0.31, 0.20	0.95	0.74, 1.22	0.687
Processed meat	−0.08	0.12	−0.32, 0.15	0.92	0.73, 1.16	0.485
Fish	0.28	0.10	0.08, 0.49	1.33	1.08, 1.63	0.006
Eggs	0.08	0.12	−0.15, 0.31	1.08	0.86, 1.36	0.498
Legumes	−0.16	0.21	−0.58, 0.26	0.85	0.56, 1.29	0.454
Cooked vegetables	0.93	0.30	0.34, 1.52	2.53	1.40, 4.57	0.002
Raw vegetables	0.26	0.13	0.01, 0.51	1.30	1.01, 1.66	0.039
Fruits	0.68	0.27	0.15, 1.22	1.98	1.16, 3.40	0.013
Cereals (e.g., spelt, barley)	0.21	0.18	−0.14, 0.56	1.23	0.87, 1.75	0.242
Whole-grain foods	0.67	0.18	0.32, 1.02	1.96	1.38, 2.78	<0.001
Nuts	0.30	0.17	−0.04, 0.64	1.34	0.96, 1.89	0.089
Tofu	0.50	0.11	0.28, 0.71	1.64	1.33, 2.03	<0.001
Dark chocolate (≥70%)	0.46	0.16	0.13, 0.78	1.58	1.14, 2.18	0.006

**Table 3 sports-13-00258-t003:** Associations between gender and individual food preferences among sport participants. Multivariable logistic regression models evaluating the association between gender (reference: female) and each food preference (dependent variable, Yes/No) among sport participants, adjusted for age and smoking status. For each food, the table reports the regression coefficient (β), standard error (SE), 95% confidence interval (CI) for β, odds ratio (OR), 95% CI for OR, and *p*-value. All 19 food preferences are included. Statistically significant results (*p* < 0.05) are highlighted in bold.

	β	SE	95% CI (β)	OR	95% CI (OR)	*p*-Value
Cow’s milk	0.20	0.11	−0.00, 0.41	1.23	1.00, 1.51	0.053
Vegetable drinks (e.g., soy milk)	−0.28	0.11	−0.49, −0.07	0.76	0.61, 0.93	0.008
Low-fat white yogurt	0.12	0.11	−0.09, 0.32	1.12	0.91, 1.38	0.274
Fresh cheeses	−0.08	0.14	−0.35, 0.19	0.92	0.71, 1.21	0.558
General meat	1.27	0.29	0.69, 1.84	3.54	2.00, 6.28	<0.001
White meat	0.67	0.22	0.24, 1.09	1.95	1.27, 2.98	0.002
Red meat	1.15	0.20	0.77, 1.54	3.17	2.15, 4.65	<0.001
Processed meat	0.72	0.17	0.40, 1.05	2.06	1.49, 2.86	<0.001
Fish	0.27	0.14	−0.01, 0.54	1.31	0.99, 1.72	0.058
Eggs	0.63	0.16	0.31, 0.95	1.87	1.36, 2.58	<0.001
Legumes	0.16	0.17	−0.17, 0.48	1.17	0.84, 1.61	0.341
Cooked vegetables	0.04	0.23	−0.42, 0.50	1.04	0.66, 1.65	0.868
Raw vegetables	0.15	0.12	−0.09, 0.39	1.16	0.91, 1.48	0.218
Fruits	0.09	0.13	−0.17, 0.35	1.09	0.85, 1.42	0.489
Cereals (e.g., spelt, barley)	0.05	0.13	−0.21, 0.30	1.05	0.81, 1.35	0.702
Whole-grain foods	0.03	0.13	−0.23, 0.28	1.03	0.80, 1.33	0.831
Nuts	0.06	0.13	−0.19, 0.31	1.07	0.83, 1.36	0.642
Tofu	0.16	0.13	−0.09, 0.41	1.17	0.92, 1.51	0.206
Dark chocolate (≥70%)	0.09	0.13	−0.17, 0.36	1.10	0.84, 1.43	0.480

## Data Availability

The data supporting the findings of this study are available from the corresponding author upon reasonable request. All data will be shared in a de-identified format to protect participant confidentiality.

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
