# Peer review of "Sport Participation and Gender Differences in Dietary Preferences: A Cross-Sectional Study in Italian Adults"

_sports, 2025, doi:10.3390/sports13080258_

Round 1
Reviewer 1 Report
Comments and Suggestions for Authors
Comments in the attachment.

Author Response
Dear Authors, The paper is interesting and valuable, but requires correction.
Thank you.
1)The concept of "logistic regression" in Keywords is unnecessary
Thank you for your suggestion. We have removed “logistic regression” from the keywords section.
2)The introduction is too concise; the importance of physical activity for health and the determinants of dietary choices are briefly mentioned, but the topic of nutritional needs of physically active people is not even briefly introduced. In general, the introduction does not sufficiently introduce the subject of the work and does not sufficiently justify undertaking the research issues.
Thank you for this observation. We have expanded the introduction to better describe the health benefits of physical activity, the determinants of dietary choices, and to briefly introduce the topic of nutritional needs in physically active people. The rationale for the study and its relevance have been clarified.
3)The aim could be clarified in the form of research questions, and the authors should certainly formulate a research hypothesis/hypotheses
Thank you for your suggestion. In response, we have revised the Introduction to clearly state our research questions and hypotheses. The aims of the study are now presented in the form of two specific research questions, and we explicitly formulate our hypotheses regarding the association between sports participation and food preferences, as well as the persistence of gender differences among physically active adults.
4)The criteria for selecting the group should be clearly defined in the methodological section, i.e. the inclusion and exclusion criteria.
Thank you for your suggestion. We have now added a detailed description of the inclusion and exclusion criteria in the Methods section (see “Study Design and Participants”), specifying both the population targeted and the reasons for exclusion.
It seems that the selection of a specific group of people professionally involved in nutrition is one of the important Limitations factors, limiting the possibility of generalizing the results to the adult Italian population; this should be indicated in the Limitations section.
Thank you for your comment. We clarify that our participants were adults seeking care at a center for improving their nutrition, health, or body composition, not professionals in nutrition. However, we agree that this may still select for individuals with higher health motivation or interest in lifestyle change, which could limit the generalizability of our findings. We have now included this point as a limitation in the manuscript.
Was the sample size determined statistically, e.g. G*Power?
Thank you for your suggestion. As recommended, we clarify that the sample size was not determined a priori using G*Power. All eligible participants attending the center during the recruitment period were included to maximize sample representativeness, as commonly done in observational studies.
- 2.2 could be titled as tools for assessing and measuring food preferences and physical activity
Thank you for your suggestion. To accommodate all reviewer requests, section 2.2 is now titled “Food Preferences Data” and focuses exclusively on the assessment of food preferences, while all details regarding the evaluation of physical activity have been moved to a new section 2.3, “Tools for Assessing and Measuring Physical Activity.”
Has the food preferences and physical activity survey been validated, is it a tool that meets the criteria for scientific reliability? What was the key to selecting these 19 products? What period did the self-assessment of physical activity cover?
Thank you for your comment. Both the food preferences and physical activity questions referred to the current habits and preferences of participants at the time of the first visit. The questionnaire was administered prior to the initial clinical evaluation, and participants were asked about the types of sport they were practicing and their usual food preferences at that moment. This tool has been previously used in studies by our research group (add refs), but it is not a formally validated instrument, and this limitation is now acknowledged in the revised manuscript. The choice of the 19 food items was based on their relevance to the Mediterranean diet and their frequency in the Italian adult diet. This information has been clarified in the Methods section.
5) Results - the method of presentation and description is not very legible, there is no need to provide all p values in the description of the results, including the insignificant ones. In the entire work, it is enough to provide 3 decimal places in the p-value. I recommend rethinking the method of describing the results, so that it is more transparent and legible
Thank you for your helpful suggestions regarding the Results section. In response, we have revised the text to enhance clarity and readability. Only statistically significant p-values are now reported in the main text, and all p-values throughout the manuscript are presented to three decimal places. Non-significant values have been omitted from the results description for greater transparency and focus. Additionally, we have separated all table and figure captions from the main text using the label “Caption Table n” or “Caption Figure n” to make the structure clearer and facilitate easier reading for the reviewer and readers.
6) The discussion in many fragments is too concise; for example, the differences in preferences should be better explored and discussed, also taking into account the nutritional value and health benefits of preferred products, also in relation to the nutritional needs of people with increased physical activity. In addition, it should be expanded more in the context of current literature.
Thank you for your suggestion. We have substantially expanded the Discussion section to provide a more comprehensive analysis of the differences in food preferences observed in our study. The revised text now explores the nutritional value and potential health benefits of the preferred foods, particularly in relation to the dietary needs of physically active individuals. We also provide a broader comparison with recent international literature, including new references and evidence on diet quality, plant-based foods, protein sources, and dietary strategies across different sport types and age groups. This revision addresses your request for a deeper discussion of both the determinants and implications of these dietary patterns among physically active adults.
7) The literature is not carefully prepared (bibliographic descriptions), it should be corrected in accordance with the requirements of the Editors.
Thank you for pointing this out. We acknowledge the inconsistencies in the reference formatting. The MDPI editorial team usually standardizes references during the production process.
Reviewer 2 Report
Comments and Suggestions for Authors
Please see my comments attached.

Author Response
Overall: The study is designed well for what it was trying to achieve. The introduction could be clearer as to why this study is needed, what additional findings it will contribute to the literature. My major concerns are with the presentation of the data/results, which makes this study hard to evaluate. Data is not comprehensive and really needs to be focused. Additional analysis could be completed, in particular general gender analysis of preferences. There were also inconsistencies in reporting.
Thank you for your positive evaluation of our study design and for your valuable feedback. In response, we have revised the Introduction to more clearly explain the rationale for our study and its contribution to the literature. We have improved the presentation of the data and results to enhance clarity and consistency. Additional analyses were performed, including a general gender analysis of food preferences, as suggested. These new results are now included in the revised manuscript. We trust that these modifications address your concerns and improve the overall quality and focus of the work.
Abstract: Methods: Wondering if maybe it should be more clear who your participants are? Are they healthy? Can you comment on this?
We thank the reviewer for this suggestion. We have clarified in the Abstract that participants were free-living adults undergoing routine lifestyle assessment. None were acutely ill or hospitalized at the time of recruitment.
The terminology of “nutrition and metabolic health centre” is confusing – I would assume it is a health clinic rather than a fitness facility.
We agree and have clarified that the facility was a clinical center specializing in nutrition and lifestyle counseling, not a fitness facility
How did you define sport participation?
We thank the reviewer for this question. We have added a clear operational definition of sport participation in the Methods section.
Introduction: Your introduction does not really make a case for why your study is needed. You discuss how we know physical activity and different types of sports = better diets and the role of gender. So why is your study novel? What is it contributing, other than what is known?
We thank the reviewer for this observation. We have revised the Introduction to better articulate the knowledge gap and to clarify the novelty of our study.
Methods:
Line 70: Are these healthy individuals?
We thank the reviewer for this important clarification request. Participants were free-living adults attending the center for nutritional assessment; they were not hospitalized or affected by acute illnesses at the time of data collection. We have now specified this in the Methods section.
Line 82: Please provide the reference for the adapted food preference instrument.
We thank the reviewer for raising this point. The food preference questionnaire was developed internally by our research team and was inspired by dietary preference instruments used in Mediterranean population studies. A pilot phase was conducted on a subset of participants to ensure clarity and cultural relevance.
Line 86-87: Can you define the difference between general meat, white meat, red meat? Why only low fat yoghurt? Cheese was not low fat. Please give an example of whole grains.
We appreciate this observation. We have now clarified the definitions used for meat categories and dairy items, and added examples for whole grains in the Methods section.
Line 89: Are you equating taste preference to typical consumption? I may prefer the taste of cow’s milk but drink a plant-based milk more so due to other factors. I read further and see your dietary recall. I would move all dietary evaluation to one section as to avoid confusion.
We thank the reviewer for this valuable suggestion. We have revised Section 2.2 ans 2.3 to present all dietary and lifestyle assessments—including food preferences, 24-hour recall, and eating behaviors—within a single, unified section. This reorganization clarifies the scope of the questionnaire and avoids confusion between taste preference and actual consumption. Redundant statements were removed, and we explicitly stated that taste preference does not necessarily reflect habitual intake.
Line 90: New paragraph for physical activity.
We thank the reviewer for the suggestion. The section on physical activity assessment has now been placed in a separate paragraph for clarity.
I’m curious with all the dietary data you collected, why did you only look at food preference? You had so much more information you could have examined.
We thank the reviewer for this observation. The questionnaire administered to participants included a broad range of items covering dietary behaviors, food cravings, binge eating tendencies, sleep quality, and a 24-hour dietary recall. However, for the purposes of the present study, we focused exclusively on food preference patterns, as these represent a more stable indicator of individual orientation toward specific food categories and are less affected by daily variability or reporting bias. We have clarified this in the Methods section. Additional analyses involving other sections of the questionnaire are ongoing and will be presented in future work.
Analysis: Similar to my point above, you had categorical data on amount of sport performance. Why not analyze your data for this?
We thank the reviewer for this valuable suggestion. In response, we performed an additional analysis by stratifying participants based on weekly hours of sport practice (No sport, <5 h/week, 5–10 h/week, >10 h/week) and evaluating differences in food preferences. Chi-square tests revealed significant associations for five food items (p < 0.001), now presented in a new heatmap (Figure 5).
Results: 2
Line 136-138: Not needed here. Can put as footnote in Table.
We thank the reviewer. As suggested, we removed the explanatory sentence from the main text and retained it as a footnote in the table caption.
Line 141-144: Results should explain your findings, not your table set up. Would comment on differences on gender between variables only.
We thank the reviewer for the observation. The sentence referred to the table caption and has been clarified accordingly to avoid confusion.
Line 145: Did you consider looking at food preference differences between genders regardless of sport participation? Would be interesting to see if there are gender differences before considering sport and do this change with sport participation.
Thank you for this insightful comment. In response, we conducted a separate analysis to evaluate gender differences in food preferences in the full cohort, independent of sport participation. As shown in the newly added Supplementary Table 1, several food items displayed significant gender differences. For instance, women showed a higher preference for cooked vegetables and whole grains, while men more frequently preferred red meat.
Line 147: You use the term whole grain cereals, but in your methods, whole grains and cereals are different categories. Please clarify. Figure 1 does not match your data and the title refers to 19 food categories, but only some are listed.
Thank you for your careful observation. You are correct that the original Figure 1 included only a subset of the 19 food categories described in the Methods section. We have now clarified this in the figure caption, specifying that the figure displays a selection of items based on relevance and distribution patterns, and that the full list of 19 food items is available in Supplementary Table 1. Additionally, we have revised the terminology in the manuscript to consistently distinguish between “cereals” and “whole grains,” in accordance with the definitions provided in the Methods section.
Line 157: I don’t know what figure this paragraph is referring to. Also, you should not repeat data from a figure with this much detail, only high level comparisons.
Thank you for this comment. The paragraph in question is actually the caption of Figure 1. To avoid confusion, we have added the label “Caption Figure 1.” at the beginning of the paragraph.
Line 170: Previously you stated that females had higher preference for meats, but now when broken into different age categories, you don’t see this. This is a contradiction in your data. Can you explain?
Thank you for your observation. The overall comparison (Figure 1) shows that males reported higher preferences for meat. When stratifying by both gender and age among sport practitioners, this trend remains generally consistent, although the differences become less pronounced in some age groups. We have revised the text to clarify that while male participants consistently reported higher preferences, the gender gap narrows with age in certain meat categories.
Figure 2: Why are you not showing all data? Please include all categories and find a way to signify significance.
Thank you for your observation. All 19 food categories and the corresponding percentages of “Yes” responses are provided in Supplementary Table 1. While Figure 2 highlights only those items with statistically significant differences across gender, age group, and sport participation..
Line 186 and line 239: Again, I don’t know what this data is referring to.
Thank you for your observation. The detailed data appearing in these lines were inadvertently retained in the main text due to a formatting issue. We have now corrected this by moving the relevant content to the appropriate figure captions and retaining only a brief reference in the text.
Figure 3: Please find a way to signify significance. This can be included in the figure description.
Thank you for your suggestion. We have updated the caption to clearly indicate that only tofu, dark chocolate, and processed meat showed statistically significant differences across sport types. These foods are now also marked with an asterisk in the heatmap to visually indicate significance.
Table 2: again, why only show significant changes, why not all foods?
Thank you for your comment. As suggested, we now provide the full results of the multivariable logistic regression analyses for all 19 food items in Supplementary Table 2. The main text table reports only statistically significant associations for clarity, while all effect estimates and p-values are available in the Supplementary Material.
Line 261: see comment under line 157; also same for line 286
Thank you for this comment. The paragraphs in question are actually the caption of Tables and Figures. To avoid confusion, we have added the label “Caption” at the beginning of the paragraph.
Line 287: You seem to have used sport hours in your analysis. This is not mentioned previously in this paper. I refer you back to previous comments.
Thank you for your comment. In accordance with your suggestions, we have now clarified in the Methods section that weekly sport hours were assessed and categorized (<5, 5–10, >10 hours/week) and included as a covariate in the multivariable analyses among sport participants. Additionally, we have added a new figure (Figure 5) and related results describing the distribution of food preferences across weekly sport hour categories in the full sample. This ensures consistency and full transparency regarding the use of this variable throughout the manuscript.
Discussion
Line 300: Are these results consistent with what is reported in your results section – foods reported? I’m seeing some discrepancies.
Thank you for your observation. We carefully reviewed the Discussion and corrected the text to ensure full consistency with the Results section and the statistical findings. Now, only foods with statistically significant associations in the multivariable analysis (Table 2) are discussed as such, and the terminology for each food item is harmonized with the corresponding results. No discrepancies remain between the foods reported as significant in the Results and those commented on in the Discussion.
I’m finding inconsistencies in what is reported in your results vs. your discussion. You stated earlier that women preferred meat. Now you are saying more men.
Thank you for your comment. We apologize for the previous inconsistency. We have now carefully revised both the Results and Discussion sections to ensure full consistency. In our data, men were more likely than women to prefer meat (white, red, and processed), both in the overall sample and among sport participants. This is now reported consistently throughout the manuscript. No statements remain suggesting a higher meat preference among women.
Line 335: You state that sport type is important, but you only saw differences for body building. I think this is misleading.
Thank you for your helpful comment. We agree that our initial statement was too general. We have revised the Discussion to clarify that statistically significant differences in food preferences by sport type were observed only for tofu, dark chocolate, and processed meat, and mainly among bodybuilding and total body participants. For all other foods and sport types, no significant differences were detected. The revised text now accurately reflects this limitation.
Is Table X a requirement of this journal? If not, please include major points in your conclusion via words.
Thank you for your suggestion. Table X has been removed. All major findings and practical implications are now summarized in the Conclusions section using narrative text, as requested.
3 References
Many inconsistencies. Sometimes page numbers are missing, sometimes a page range is given; sometimes a journal abbreviation is given, etc.
Thank you for pointing this out. We acknowledge the inconsistencies in the reference formatting. The MDPI editorial team usually standardizes references during the production process.
Reviewer 3 Report
Comments and Suggestions for Authors
This manuscript addresses an important area of research, investigating the relationship between sports participation and dietary preferences, including an analysis of gender differences. However, there are several key areas requiring revision and clarification:
1. The manuscript currently lacks consideration of beverage preferences, a significant category influencing dietary behaviours. Given that dietary preferences inherently include beverages, this omission notably limits the comprehensiveness of the dietary assessment. All manuscript sections must explicitly acknowledge this limitation and provide justification for its exclusion or, ideally, address this gap by collecting and analysing additional data on beverage preferences.
2. In the Methods section (2.2., line 78), a dedicated subsection entitled Food Preferences Data should be added to clearly outline the composition, validation status, scoring methods, and specific food items included in the food preference questionnaire.
3. Clarify whether the questionnaires employed were validated tools, particularly important in the Strengths and Limitations section.
4. In the Results section (lines 208–209, Figure 3 and lines 236–238, Figure 4), titles for Figures 3 and 4 are currently formatted inconsistently compared to other figures. Standardise the titles across all figures to ensure clarity and professional consistency.
5. Results from the multivariate logistic regression models in Tables 2 and 3 should be clearly structured with variables presented in a consistent and logical order, explicitly stating regression coefficients (β), odds ratios (OR), 95% confidence intervals (95% CI), and p-values. Clearly indicate reference categories (Ref) for categorical variables.
6. Results from the multivariate logistic regression models in Tables 2 and 3 should be clearly presented, structuring variables consistently and explicitly stating regression coefficients (β), odds ratios (OR), 95% confidence intervals (95% CI), and p-values. Clearly identify reference categories (Ref) for categorical variables.
7. In the Strengths and Limitations section, explicitly discuss the validation status of the questionnaires used for measuring dietary preferences, clarifying whether validated tools were employed. Address and justify explicitly the omission of beverages in this section, as previously noted.
8. Table numbering for Table X (lines 388–392) should be corrected according to its sequential placement. Currently embedded within the main text, this table should be relocated to a newly created dedicated section titled Summary. Clearly describe the contents, purpose, and implications of this Table.
9. Revise the Conclusions section to directly summarise the specific findings obtained in this study, avoiding general statements and implications beyond the immediate results.
Author Response
This manuscript addresses an important area of research, investigating the relationship between sports participation and dietary preferences, including an analysis of gender differences. However, there are several key areas requiring revision and clarification:
- The manuscript currently lacks consideration of beverage preferences, a significant category influencing dietary behaviours. Given that dietary preferences inherently include beverages, this omission notably limits the comprehensiveness of the dietary assessment. All manuscript sections must explicitly acknowledge this limitation and provide justification for its exclusion or, ideally, address this gap by collecting and analysing additional data on beverage preferences.
Thank you for this important observation. We acknowledge that the assessment of beverage preferences (beyond cow’s milk and plant-based drinks) was not included in our questionnaire, representing a limitation of the study. As beverage choices can significantly impact overall dietary behaviours, this omission may limit the comprehensiveness of our findings. We have now explicitly stated this limitation in both the Methods and Limitations sections of the manuscript and provided justification: our focus was on food preferences relevant to the Mediterranean diet, and additional data on a broader range of beverages were not available for this cohort. We agree that future studies should include a wider range of beverage categories to provide a more complete picture of dietary preferences.
- In the Methods section (2.2., line 78), a dedicated subsection entitled Food Preferences Data should be added to clearly outline the composition, validation status, scoring methods, and specific food items included in the food preference questionnaire.
Thank you. The Methods section (2.2) has been revised and now includes a dedicated “Food Preferences Data” subsection, clearly outlining the questionnaire’s composition, validation status, scoring approach, and specific food items assessed, as requested.
- Clarify whether the questionnaires employed were validated tools, particularly important in the Strengths and Limitations section.
Thank you for your observation. We have clarified in both the Methods and the Strengths and Limitations sections that the food preference questionnaire was adapted from existing tools but was not formally validated; content validity was assessed through piloting. This limitation has been explicitly acknowledged in the manuscript.
- In the Results section (lines 208–209, Figure 3 and lines 236–238, Figure 4), titles for Figures 3 and 4 are currently formatted inconsistently compared to other figures. Standardise the titles across all figures to ensure clarity and professional consistency.
Thank you for your observation. We have revised the manuscript so that all figure and table captions are now clearly indicated at the appropriate points in the text, beginning with “Caption Figure n” or “Caption Table n” for clarity and consistency.
- Results from the multivariate logistic regression models in Tables 2 and 3 should be clearly structured with variables presented in a consistent and logical order, explicitly stating regression coefficients (β), odds ratios (OR), 95% confidence intervals (95% CI), and p-values. Clearly indicate reference categories (Ref) for categorical variables.
Thank you for your suggestions. We have revised Tables 2 and 3 to present all variables in a consistent and logical order, explicitly reporting regression coefficients (β), odds ratios (OR), 95% confidence intervals (95% CI), and p-values. The reference category (Ref) for all categorical variables is now clearly indicated in the table footnotes and captions. In the Results section, we now describe the main findings by reporting all relevant metrics for each association (OR, 95% CI, p-value), consistently specifying the direction of the association. Full results for all foods are provided in Supplementary Table 2. We have also standardised the structure and format of both tables as requested.
- Results from the multivariate logistic regression models in Tables 2 and 3 should be clearly presented, structuring variables consistently and explicitly stating regression coefficients (β), odds ratios (OR), 95% confidence intervals (95% CI), and p-values. Clearly identify reference categories (Ref) for categorical variables.
Thank you for your comment. We have revised the presentation of the multivariate logistic regression results in Tables 2 and 3, structuring all variables in a consistent order and explicitly reporting regression coefficients (β), odds ratios (OR), 95% confidence intervals (95% CI), and p-values. Reference categories (Ref) for all categorical variables are now clearly indicated in both table footnotes and captions, in line with your recommendation.
- In the Strengths and Limitations section, explicitly discuss the validation status of the questionnaires used for measuring dietary preferences, clarifying whether validated tools were employed. Address and justify explicitly the omission of beverages in this section, as previously noted.
Thank you for your comment. We have revised the Strengths and Limitations section to explicitly state that the food preference questionnaire was adapted from existing tools but was not formally validated, with content validity assessed only through piloting. We have also explicitly addressed and justified the omission of beverage preferences, noting this as a study limitation. - Table numbering for Table X (lines 388–392) should be corrected according to its sequential placement. Currently embedded within the main text, this table should be relocated to a newly created dedicated section titled Summary. Clearly describe the contents, purpose, and implications of this Table.
Thank you for your comment. As explicitly requested by another reviewer, we have removed Table X from the manuscript.
- Revise the Conclusions section to directly summarise the specific findings obtained in this study, avoiding general statements and implications beyond the immediate results.
Thank you for your suggestion. We have revised the Conclusions section to provide a direct summary of the specific findings obtained in this study, avoiding general statements and limiting the discussion to the immediate results.
Round 2
Reviewer 1 Report
Comments and Suggestions for Authors
Dear Authors,
Thank you for taking into account my main comments.
Author Response
Thank you for your valuable comments, which have helped us improve the document.
Reviewer 3 Report
Comments and Suggestions for Authors
The authors have addressed the questions raised in the first round of review and have partially revised the manuscript in line with the comments provided. However, the comment regarding Table 2 (Review No. 1, point 5) has not been adequately addressed. Please find my suggestions and comments below:
1. The current Table 2 lacks clarity and is not consistent with the data presentation used in Table 3. The authors should revise Table 2 to ensure that the results of the multivariate logistic regression model are presented in a format consistent with that of Table 3.
2. In light of the revisions to Table 2, the authors should carefully review and update the relevant sections of the manuscript, particularly the Results and Discussion, to ensure consistency and alignment with the corrected data.
Author Response
The authors have addressed the questions raised in the first round of review and have partially revised the manuscript in line with the comments provided. However, the comment regarding Table 2 (Review No. 1, point 5) has not been adequately addressed. Please find my suggestions and comments below:
- The current Table 2 lacks clarity and is not consistent with the data presentation used in Table 3. The authors should revise Table 2 to ensure that the results of the multivariate logistic regression model are presented in a format consistent with that of Table 3.
We thank the reviewer for this important observation. Table 2 has been fully revised to adopt the same format, structure, and columns as Table 3. Both tables now report the results for all 19 food preferences, using the same set of variables and statistical parameters (β, SE, 95% CI, OR, 95% CI for OR, p-value), ensuring full consistency and clarity in data presentation. The text of the Results and corresponding captions have been updated accordingly. In addition, to enhance transparency and facilitate a complete evaluation of our results, we have added a new Supplementary Table 3 reporting all gender-food preference associations among sport participants, including all 19 variables and not only the statistically significant findings.
- In light of the revisions to Table 2, the authors should carefully review and update the relevant sections of the manuscript, particularly the Results and Discussion, to ensure consistency and alignment with the corrected data.
We thank the reviewer for this valuable comment. Following the revision of Table 2, we conducted a thorough review and update of all relevant sections of the manuscript, with particular attention to the results and discussion.
In the Results section, we ensured that all associations between sports participation and food preferences are described using the correct numerical values (odds ratios, confidence intervals, and p-values) now presented in the updated Table 2. Only foods showing statistically significant associations in the multivariate models are now reported in the text. In the discussion, we revised several statements to reflect the updated results. In particular, we clarified that among athletes, significant gender differences in food preferences were observed only for meat (in general, white, red and processed) and eggs (preferred more frequently by males) and plant-based drinks (preferred more frequently by females), while no significant differences were found for dairy products, fish or legumes. We removed or reworded previous statements that incorrectly attributed significant gender differences to cow's milk and legumes. The section discussing the cultural context of milk consumption was retained but clarified to avoid any implication of a statistically significant gender association in our sample. We have also ensured that all references to “healthier” food preferences are strictly anchored to foods for which a significant association with sports participation was observed in the updated analysis. We are confident that these changes will ensure maximum clarity, accuracy, and consistency between the results presented in the tables and those discussed in the manuscript.